# A Mechanical Modelling and Simulation Method for Resolving PIM Problems in Antennas

**DOI:** 10.3390/s22010294

**Published:** 2021-12-31

**Authors:** Chen Chen, Yangyang Gu

**Affiliations:** 1Institude of Mechanical and Electronic Engineering, Northeast Forestry University, Harbin 150040, China; 2Institude of Electronic and Information Engineering, Soochow University, Suzhou 215006, China; yygu0314@stu.suda.edu.cn

**Keywords:** passive intermodulation (PIM), sine vibration, harmonic simulation, mechanical finite element simulation

## Abstract

Passive intermodulation (PIM) generated from antennas is a nonlinear distortion phenomenon and causes serious problems to communication quality. Traditional radio frequency (RF) solutions focus on testing the final product to find the PIM source. However, it cannot solve the stability of PIM after the antenna is vibrated. This paper introduces a new method to improve the stability of PIM in the design phase. By studying the mechanism of PIM generation, a simulation method is proposed in this paper by applying mechanical finite element simulation and simulating the structural design of the device under test. Then, the stress at the PIM source is reduced, thereby the PIM stability of the product is improved. This paper adopts this method by studying a typical product, finding the root cause that affects the product PIM magnitude and stability, and optimizing its design. The PIM value of the new scheme is stable by making a prototype and testing. The method provided in this article can effectively improve product development efficiency and assist designers in avoiding the risks of PIM before the product’s manufacturing.

## 1. Introduction

Passive intermodulation (PIM) is generated when two carrier signals pass through a nonlinear passive component at the same time [1,2,3,4,5,6,7]. In mobile communication networks, PIM distortion may reduce the performance of a multi-channel wireless communication system. PIM will also cause the noise level to increase, thereby weakening the sensitivity of the receiver and eventually causing the user’s communication quality to deteriorate and the communication to be halted in severe cases.

A poor quality antenna will cause PIM. PIM often happens in some passive components of antennas which are generally considered as linear devices, such as the phase shifter, dipole, cables, and connecters. PIM issues are usually caused by weak metal-to-metal contact through the signal path [1]. They are mainly manifested in bad solder joints and poor torque of fasteners in the transmission path [2].

In practice, the base station antennas are installed on the top of the base station. These antennas bear both wind vibration and vibration from the base station. Since base stations are expected to have a useful lifetime of more than ten years, the industry generally refers to the base station antenna standards by the NGMN Alliance (BASTA) using the sine sweep vibration test, shock and bump test, and broadband random vibration test to accelerate fatigue experiments on base station antennas. The worst case among these tests is the sine sweep vibration test according to years of test results. After the sine vibration test, if the PIM value drops below the threshold, it is considered that the product cannot reach the service life.

The source where PIM is generated can be identified by various tests. At present, the main test methods for identifying PIM sources are the reverse PIM test, distance-to-PIM test, high-power test, and vibration modulation method [8,9,10]. The tests of these methods are very efficient, and each has a range of applicability. Among them, the distance-to-PIM method can roughly test the approximate position of the PIM source in the signal path, and it is also used by some equipment manufacturers to detect the PIM source of the product. At the same time, it helps engineers to repair the products. High-power tests can more accurately identify products with poor PIM and avoid the “false negatives” problem [11]. The vibration modulation method aims to detect the PIM source on the device under test through specific frequency vibration. The vibration of the probe sonde will cause amplitude modulation at the same frequency as the vibration setting frequency, thereby amplifying the PIM source signal, making the PIM source easier to be recognized.

These methods have limitations; that is, the source can be detected only after the equipment is produced. For example, the distance-to-PIM method can quickly find the PIM source after the product completes the vibration test, but it cannot visually see the structure failure mechanism, and there is no reference for changing the design [12]. The vibration modulation method can impose acoustic vibration on the transmission link according to the designer’s experience, thereby amplifying the effect of the PIM source [13]. The limitation of the vibration modulation method is that it can only vibrate the local structure and cannot fully reflect the actual vibration response of the device.

For the time being, in order to maintain the quality of the antenna and improve the design, it will go through a PIM test—source identification—optimization of design—prototype manufacturing—PIM test cycle. This cycle is illustrated in Figure 1.

There are two obvious shortcomings of the traditional workflow. On the one hand, it all depends on the experience of the engineer, who seldom knows what happens inside the antenna while doing a sine sweep vibration. Sometimes the root cause is at a sub-component, such as phase shifter, which the engineer has never considered. On the other hand, to make a prototype is very time consuming. If the improved prototype fails, the development of product may pend for a long time, or even worse, the development of the product may be canceled.

The main reason why the PIM problem is difficult to solve is the inevitable large number of solder points in the antenna transmission link, the metal connection points, and the assembly stresses generated by the assembly. Moreover, PIM has the characteristic of being time-varying because the nonlinearity depends on the life of the component, and the fastener connection will become weaker after a period of storage and transportation. Since outdoor wireless devices in practical applications will be fixed at high places for a long time, PIM will be generated due to the wind vibration. In the actual design, accelerated aging tests are carried out through sine vibration and random vibration to predict the reliability of the design [14,15,16].

In order to overcome the complex manufacturing, test, design, and test cycle, researchers attempt to predict the PIM source by simulations. Some of them focus on electrical simulation by simulating the generation of PIM source [17,18,19]. One of the studies built a 3D model of the metal-to-metal contact, as well as the electromagnetic environment inside the device, then the simulation result showed the nonlinear current at the receiving antenna [20,21,22,23]. These simulation studies on PIM simulate the generation and trend of PIM, but do not propose any method to improve the design so as to either reduce or eliminate PIM.

In order to address the problem of PIM in antennas, a method based on finite element simulation to solve the PIM problem is proposed in this paper. A mechanical harmonic response simulation method is proposed to improve the reliability of equipment PIM from a mechanical point of view to make up for the deficiencies in the above methods.

Firstly, the mechanical finite element simulation is used to simulate the structural design of the equipment on the side. The finite element method (FEM) is a numerical technique for analyzing engineering designs. FEM is accepted as the standard analysis method due to its generality and suitability for computer implementation. FEM divides the model into many small pieces of simple shapes called elements, effectively replacing a complex problem by many simple problems that need to be solved simultaneously. Elements share common points called nodes. As the Figure 2 shows, the process of dividing the model into small pieces is called meshing. The response at any point in an element is interpolated from the response at the element nodes. Each node is fully described by a number of parameters depending on the analysis type and the element used.

Then, the maximum stress point under the resonance frequency response [24,25,26] is found. After that, according to the analysis of structural harmonic response deformation and stress distribution, structural optimization design is carried out. The purpose of this structural optimization design is to reduce stress at the current concentration. At the same time, the main components are distributed in the maximum deformation area far away from the resonance frequency point to avoid large relative deformation between the components on the transmission path during the vibrator process [4,27,28]. This relative deformation results in damage to the connections on the electrical transmission path, creating a PIM risk [29,30,31,32].

The proposed method is implemented in a SolidWorks simulation environment and tested on PIM test equipment from Summitek. From comparing the simulation results and test results on purpose-made antennas, it can be seen that the proposed simulation method is very effective, and a new design based on the simulations can significantly reduce PIM.

This paper is organized as follows: Section 2 describes the mathematical modelling of PIM in antennas. Simulations and performance evaluation are presented in Section 3. Tests and validation are presented in Section 4. This paper is concluded in Section 5.

## 2. Modelling of PIM in Antennas

Figure 3 shows the connection of the individual components and transmission path of a typical antenna.

On the transmission path, PIM sources can be divided into three categories: joint connections, welding points between components, and poor contact between components and reflectors.

All three types of bad contact occur at the connection of the component to the cable so we needed to express the mathematical model of the PIM generated on the transmission path. A point source model which accounts for the two PIM signals generated by multiple intermodal sources was proposed by Deats and Hartman [17]. As everyone knows, cable assembly includes two kinds of PIM sources, forward and reflected. Two constant power transmission signals are input to the cable assembly at Port 1. The forwarded PIM and the reflected PIM are measured at Port 1 and Port 2, respectively. As shown in Figure 4, it was assumed that only the connector is the source of PIM generation. The cable did not generate PIM signal significantly. It was also assumed that the connectors’ loss was none and all connectors were the same. In addition, once this PIM was detected, starting from the origin, there were thrusters of equal strength in both directions. We also assumed that different connectors shared the same characteristics of PIM.

In order to approximate realistic assumptions, a simply optimized model, loss and group delay cables included, was applied, as shown below:(1)$${H(\omega )=\alpha e}^{-j\omega T}{=\alpha e}^{-j\beta l}$$
where α stands for coaxial cable voltage in transmission, β stands for the number of waves related to the signal through the cable, *T* is the delay time of transmission and *l* is the length of cable, and ω is the angular frequency.

It should be noted that the loss of the connector can also be added in the sum of transmission loss. Usually, the most serious components are the third-order intermodulation terms at frequencies which can belong to the band of wanted frequencies. The filter cannot filter out these frequencies once the components of these frequencies are close to the fundamental frequencies. From the above assumptions, it can be deduced that the third-order PIM voltage of the first cable assembly connector was:(2)$$v_{1\;}{=\sigma e}^{{(2j\omega }_{1}{-j\omega }_{2})t}{=\sigma e}^{{j(2\omega }_{1}{-\omega }_{2})t}{=\sigma e}^{{j\omega }_{3}t}\;$$
where *σ* is the PIM coefficient for the connectors, *ω*_1_ and *ω*_2_ are the frequencies of carrier number 1 and 2, and *ω*_3_ is the frequency of the third-order PIM response. In this article, we only consider third-order PIM products, and higher order products can be obtained in the same way.

The PIM response of the second connector was:(3)$$v_{2}{=v}_{1}{H(\omega )\alpha }^{\gamma_{3}}{=\sigma \alpha }^{{1+\gamma }_{3}}e^{-j\beta l}e^{{j\omega }_{j}t}\;$$
where *v*_2_ is the third PIM response voltage from the second connector and *γ*_3_ is the re-growth rate of the third PIM product by input power.

In the case of two series connection, the total forward and reflected PIM voltage wave can be expressed as:(4)$$v_{2}^{-}{=v}_{1}{+v}_{2}{H(\omega )=\sigma e}^{{j\omega }_{3}t}{(1+\alpha }^{{1+\gamma }_{3}}e^{-j2\beta l})$$
(5)$$v_{2}^{+}{=v}_{1}{H(\omega )+v}_{2}{=\sigma e}^{{j\omega }_{3}t}e^{-j\beta l}{(\alpha +\alpha }^{\gamma_{3}})\;$$
where *v*^−^ and *v*^+^ are forward and reflected PIM voltage.

It was assumed that *θ* = 2βl, because it can be treated as the electrical length of the cable, and *γ*_3_ = 3. Equations (4) and (5) can be expressed as Equations (6) and (7):(6)$$v_{2}^{-}{=\sigma e}^{{j\omega }_{3}t}{(1+\alpha }^{4}e^{-j\theta })\;$$
(7)$$v_{2}^{+}{=\sigma e}^{{j\omega }_{3}t}e^{-j/2\theta }{(\alpha +\alpha }^{3})$$

In addition, the model of Figure 4 includes the magnitude of the forward and reflected PIM in case of a two-stage series connection, as shown in (8) and (9), respectively:(8)$${|v}_{2}^{-}|=\sigma \sqrt{{{(1+\alpha }^{4}\cos\theta )}^{2}{+(\alpha }^{4}{\sin\theta )}^{2}}$$
(9)$${|v}_{2}^{+}{|=\sigma (\alpha +\alpha }^{3})$$

In a communication scenario, many components are deployed in a transmission path, such as the connector, phase shifter, and power divider. This was simulated as a model with n-stage series connections, as illustrated in Figure 5. As Figure 5 shows, the components were cascaded; we assumed that the insertion loss of the cable was the same.

In the case of n-stage series connections, we found that the expressions of the total forward and reflected PIM voltage waves were:(10)$$v_{n}^{-}{=\sigma e}^{{j\omega }_{\omega }t}\overset{n}{\underset{k=1}{\sum }}\alpha^{4\left(k-1\right)}e^{-j\left(k-1\right)\theta }\;$$
(11)$$v_{n}^{+}{=\sigma e}^{{j(\omega }_{3}t-1/2\theta )}\alpha^{n-1}\overset{n}{\underset{k=1}{\sum }}\alpha^{2(k-1)}\;$$

So, the series connection factor for the n-stage series could be expressed in dB as follows:(12)$$\begin{array}{c} S_{n}^{-}=10\log(\frac{\frac{1}{2}{|v}_{n}^{-}{|}^{2}}{\frac{1}{2}{|v}_{1}{|}^{2}})\;\\ =10\log({(\overset{n}{\underset{k=1}{\sum }}\alpha^{4(k-1)}\cos((k-1)\theta ))}^{2}\\ +{\left(\overset{n}{\underset{k=1}{\sum }}\alpha^{4(k-1)}\sin((k-1)\theta )\right)}^{2}) \end{array}$$

### Mechanical Harmonic Model

Antenna structures are described in the physical domain by their mass, stiffness, and damping properties. In a typical way, the equations of motion for a discrete dynamic subsystem can be written as:(13)$$\left[M\right]\left\{\ddot{x}\right\}+\left[C\right]\left\{\dot{x}\right\}+\left[K\right]\left\{x\right\}=\left\{F\left(t\right)\right\}$$
where [M], [C], and [K] are the mass, damping, and stiffness matrices of the system {*x*} and *F*(*t*) are the displacement response vector and the excitation force vector at each point of the system, respectively.

Assuming that the structure of the antenna is free vibrating and damping is ignored, F(t)=0 and [C]=0. When harmonic vibrations occur, this equation can be deduced as:(14)$$-\omega^{2}\left[M\right]+\left[K\right]=0.\;$$
where ω is the natural frequency of the antenna structures. So, the natural frequency is determined by the stiffness matrix [*K*] and the mass matrix [*M*] of the antenna structure. Therefore, increasing the first order frequency can improve the structural stiffness of the antenna.

In the later harmonic simulations, it was only necessary to focus on the first-order response frequency and to see how the antenna structure was stressed at the first-order response frequency. The first-order response frequency could then be increased by optimizing the structure to achieve a lower PIM value for the antenna.

## 3. Simulations of PIM in Antennas and Performance Evaluation

After modeling, a new flowchart was designed to implement the models in a platform for simulations and the design of the antennas based on simulation, which is illustrated in Figure 6. First, the harmonic response simulation of the entire antenna structure was carried out to determine the force environment of the components as a reference for the subsequent static simulation. The reason for this was that harmonic response simulation consumes a lot of resources and is not suitable for orthogonal experiments of multiple schemes on the antenna level. In parallel, transmission path analysis were done. The purpose of transmission path analysis was to recognize the crucial parts of the antenna. Then, according to the results of the harmonic response simulation, the static simulation boundary conditions were designed, and the orthogonal simulation analysis was performed to obtain the optimal solution. After that, the result was compared with the original design. If the result was better than the original design, the design was manufactured as the prototype and a sine sweep vibration was carried out to ensure the quality. The last step was to test the PIM level to make sure the design was qualified.

The simulation steps of antenna structure harmonic response were as follows:

The first step of the simulation was to establish a simplified simulation model of the antenna. The principle of establishing the simplified model was to eliminate the details and retain only the parts that reflect the stiffness and mass. Figure 7 illustrates an example of simplifying a phase shifter plate. In the model of simulation, there were holes for supporting phase shifter, as well as features to insert some clips. The purpose of simplifying the model was to accelerate the simulation process and save the resource of computing.

Other types of entities in the antenna assembly needed to be simplified. For parts such as PCB and cables that have little contribution to stiffness, mass points or distributed masses were directly used to replace them. For example, if the center of mass of the radiator was far away from the contact surface, the remote mass point was used. In actual products, cables usually use cable ties or clips, so the cables only needed to be treated as a distribution mass without increasing the damping.

The next step was to set up the contact relationship. The SolidWorks harmonic response module has two limitations. On the one hand, the material properties can only be linear, which means that the nonlinear material properties are invalid in the SolidWorks harmonic response module; on the other hand, the contact setting can only set the bond type. Contacts, such as frictionless contact and other types of contacts, cannot be made. When setting the contact relationship, it was noted that if the contact form between two parts was a frictionless contact or a friction contact in static mechanical simulation then this contact needed to be deleted in the harmonic response simulation.

The next step was to set the mesh size of the part. The principle of setting the mesh size is to ensure that the contact surfaces share similar mesh sizes and types.

Next, was the setting of boundary conditions. The setting of boundary conditions mainly includes the setting of excitement points. The form of excitation is a constant acceleration, and the magnitude of excitation is equal to the actual vibration test. The value of the damping ratio is usually 0.05 in this type of simulation. In particular, according to the requirements of the BASTA experimental standard, an acceleration sensor should be attached to the product to measure the first-order resonance frequency response, and a dwell vibration experiment should be carried out at the resonance frequency point.

The isometric view of the model below shows the simulation boundary condition settings in Figure 8. There are green fixed points to generate sine sweep frequency vibration excitation.

Although the time required to solve the simulation varied depending on the complexity of the antenna, a relatively accurate result could usually be obtained within a few hours. After harmonic simulation of the overall structure of the antenna, it was found that, in the frequency range of 5–200 Hz, the deformation at the first-order natural frequency was as shown in Figure 9 below. According to harmonic simulation result, the stress on the phase shifter assembly (PSA) plate was caused by deformation of the reflector. In order to reduce the deformation of the reflector, we used diplexers as structure components which are shown as Figure 10b. Compared with the original design shown in Figure 10a, the improved design connects the side wall of the reflector. As a result, the diplexer was not only the load in vibration, but also contributed the stiffness to the whole structure.

Then, the comparison simulation was done for next step. In this stage, the simulation result of the improved design was compared with the original one. The comparison results are shown in the Table 1.

The deformation at the crucial location was also checked after simulation. Here, the comparison of deformation of reflectors at the crucial location before and after are shown in Figure 11 and Table 2.

The following results were obtained after simulation:

First, the maximum deformation occurred at the first-order resonance frequency. Second, the maximum deformation of the phase shifter plate occurred at the first-order resonance frequency as well.

Reducing the swing amplitude of the phase shifter in the vibration state reduced the damage of vibration to the welding joint, thereby reducing the risk of nonlinear signals on the transmission path. Therefore, by optimizing the scheduling of fixed points, the plate’s stiffness was enhanced in a way that reduced the amplitude of the phase shifter, which improved the PIM value.

It can be seen from the simulation results that, by adding the ribs, increasing the thickness of the plate, and increasing the strength of the cable clamp, the stiffness of the entire phase shifter can be effectively increased, and the stiffness of the entire antenna can be improved to a certain extent as well.

However, the improvement is not obvious due to the increase in plate thickness. So, increasing the ribs and strengthening the cable clamp simultaneously is a better idea.

### Static Simulation

Harmonic simulations provided a comprehensive simulation of the internal structure of the antenna during vibration. However, it was not possible to simulate the assembly stress at the PCB solder joints due to the bending of the coaxial cable during the assembly process. Therefore, this paper used static analysis to simulate the process of coaxial cable bending in order to evaluate the impact of the original design and the improved design on the PIM. As can be seen from the previous analysis, the solder joint is where the current is concentrated, so the greater the assembly stress on the solder joint, the greater the risk of the corresponding PIM at the solder joint and the worse the PIM value. So, the lower the stress near the weld joint after the static analysis is obtained, the lower the PIM risk and the better the PIM value that can be achieved at the same time.

The more concentrated the power distribution of the part is, the greater the impact is on PIM. The phase shifter part is the most concentrated part of the power. The signal is divided into multiple channels after passing through the phase shifter. At the same time, in the source design, the PIM value will be better after welding a phase shifter, so it is assumed that the design defect is at the phase shifter. If the assumption is wrong, the PIM value should not be reduced to the same degree after improvement.

In this article, an example will be given to demonstrate this method.

The two phase shifters to be compared are shown in Figure 12. The simulation compared the difference in internal stresses during coaxial cable bending for two phase shifters with different distances from the fixed point.

When the antenna is assembled and wired, the cable gives a force to the cable clamp, which drives the phase shifter board to deform it. Eventually, the force is transferred to the solder joints on the PCB. The unknown quantity at this point is the magnitude of the force generated on the cable clamp when the cable is bent. In this paper, an experiment was designed to measure the force generated when bending a cable. The experiment is shown in Figure 13.

The coaxial cable type was PTS1-50 and is widely used as an input cable for phase shifters in antenna systems. The cable was pressed until it started to bend and then the amount of force was displayed on the sensor’s screen. The force value was stabilized at 20.9 N by performing five tests on different PTS11-50 coaxial cables.

Then, the simulation model was built up. The material setting is listed in Table 3.

Figure 14 illustrates the boundary condition of the original design and the improved design. The boundary conditions were set by fixing the two sides of the phase shifter plate and applying 20.9 N force on the four places of the cable clamps. As per Saint-Venant’s principle, the difference between the effects of two different but statically equivalent loads became very small at sufficiently large distances from load; only one set of the phase shifter was built up to save the compute resource. The original design and the improved design had a similar structure, with the only difference being the distance between the fixed points. The fixed points of the original design were on the inside of the cable clamps and close to the phase shifter, and the fixed points of B were on the outside of the clip clamps and away from the phase shifter. Compared to the original design, the improved design eliminated the cantilever structure between the cable clip and the phase shifter.

First, the 16 phase shifters were assembled and the input cables were bent once to test the PIM values. The PCB was then left unchanged, the plate was replaced with a modified solution, the solder joints were re-soldered and the input cables were bent again to test the PIM values. This method ensured that the PCB was as consistent as possible in both cases. The simulation results are shown in the Figure 15 which indicated that the failure mode was probably in the solder joint section, so the solder joints were re-soldered.

## 4. Test Results and Evaluation

We assembled 16 antennas and tested the PIM values based on traditional workflow and also improved workflow. After simulations and improved design, all antennas were assembled again based on the new design.

PIM testing of base station antennas requires the use of a PIM tester. The PIM test equipment used in this study was supplied by Summitek. The SI-900 was used for the low frequency band, transmitting from 960 to 925 MHz and receiving from 915 to 880 MHz, while the SI-1800D was used for the high frequency band, transmitting from 1880 to 1805 MHz and receiving from 1785 to 1710 MHz.

Because phase shifters affect the PIM performance the most seriously, PIM test results on phase shifters are shown in Figure 16 and Figure 17. Figure 16 shows the results of the original design and improved design after bending the input cables. The bule bars indicate the results of the original design and the amber bars indicate the results after re-design. It can be seen that the PIM values were reduced by 2.75 dB in average.

Figure 17 shows the overall PIM results between the two designs. It can be seen that the performance of PIM was improved further, where the highest 10dB was achieved on antenna 3 and lowest 2 dB was achieved on antenna 10 and antenna 11. Due to the measurement tolerance of the PIM test equipment, the improvement of antenna 9 was not impressive.

To be able to highlight the comparison results better, the DUT cannot be named after its actual product code in this paper but its results can be expressed as a PIM value.

As can be seen in Figure 16, the improvement in phase shifter was very noticeable, with up to 5 dB. The improvement was also evident at the antenna level, as can be seen in Figure 17, with most antennas showing a significant improvement, while DUT8 showed a decrease, probably due to individual differences.

## 5. Conclusions

The structure of specific components in the antenna is different, and the types of key component are different in different designs, but this does not affect the use of this method. Additionally, based on the structural scheme before the simulation optimization, the PIM value decreased after the vibration experiment. According to the conclusions of previous studies, we focused on the metal-to-metal connection, the component interface, and the current density in the structural simulation. In the comparison of simulation and experimental results, it was found that reducing the maximum relative displacement of the key component phase shifter away from the first-order harmonic response had a significant effect on the stability of the PIM value of the entire system. By changing the layout, the key points on the transmission link that affected the PIM value were far away from the area with the largest warping amplitude under the first-order harmonic response. By designing plastic buffer parts, the local damping increased. Comparison of PIM values before and after improvement effectively improved the level of dynamic PIM testing.

Through the methods provided by this research, the efficiency of product research and development has been effectively improved. The harmonic response analysis method helps designers understand the internal vibration response of the product under vibration. Designers can design purposefully according to the deformation of simulation results. Simultaneously, simulation standards were formulated in the project, stipulating that the relative deformation of key components in the simulation of first-order frequency response should not be greater than a specific value. This specific value is the maximum value of the relative deformation of the key component corresponding to the similar structural product that has been tested in the harmonic response simulation. If it is greater than the maximum relative deformation of the corresponding key component in the existing simulation, then the design needs to be optimized before the standard is reached.

## Figures and Tables

**Figure 1 sensors-22-00294-f001:**
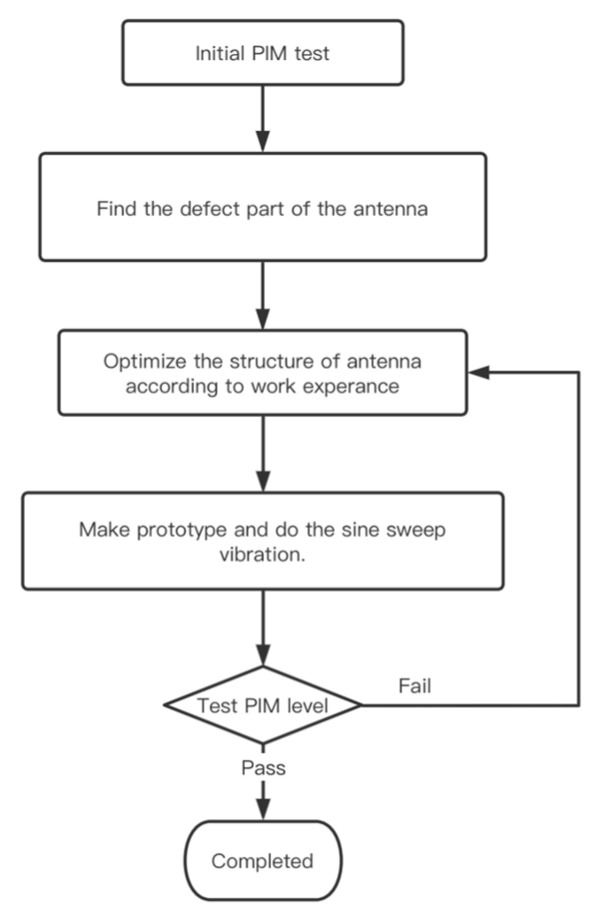
Traditional workflow of PIM improvement.

**Figure 2 sensors-22-00294-f002:**
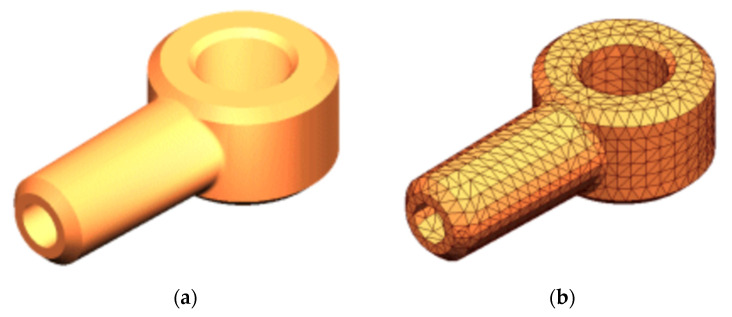
An example of finite element method (**a**) CAD model of a part (**b**). Model subdivided into small pieces (elements).

**Figure 3 sensors-22-00294-f003:**
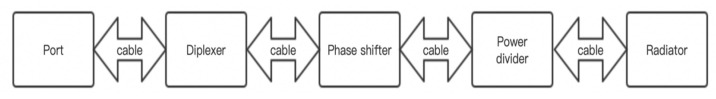
Transmission path of the typical antenna.

**Figure 4 sensors-22-00294-f004:**
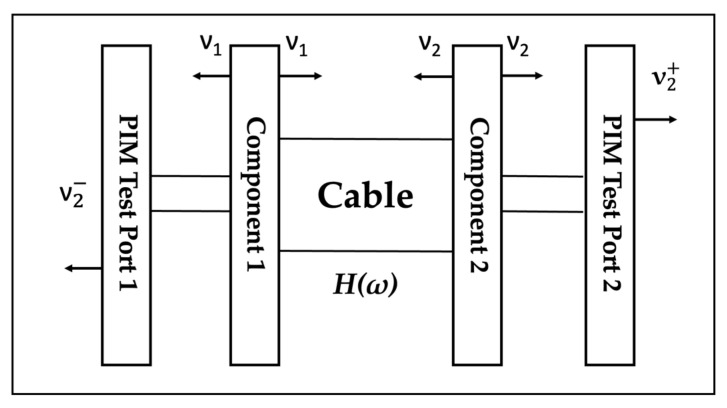
PIM response model of point source.

**Figure 5 sensors-22-00294-f005:**
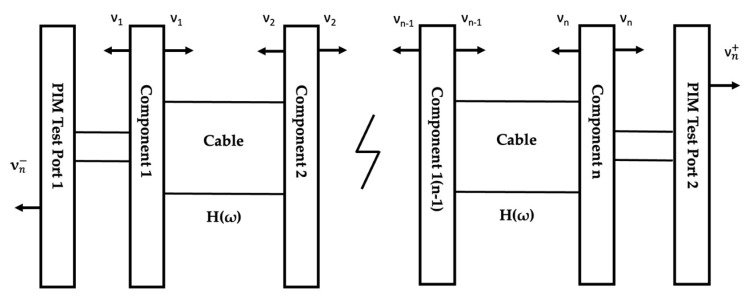
The general model for n−stage series connection.

**Figure 6 sensors-22-00294-f006:**
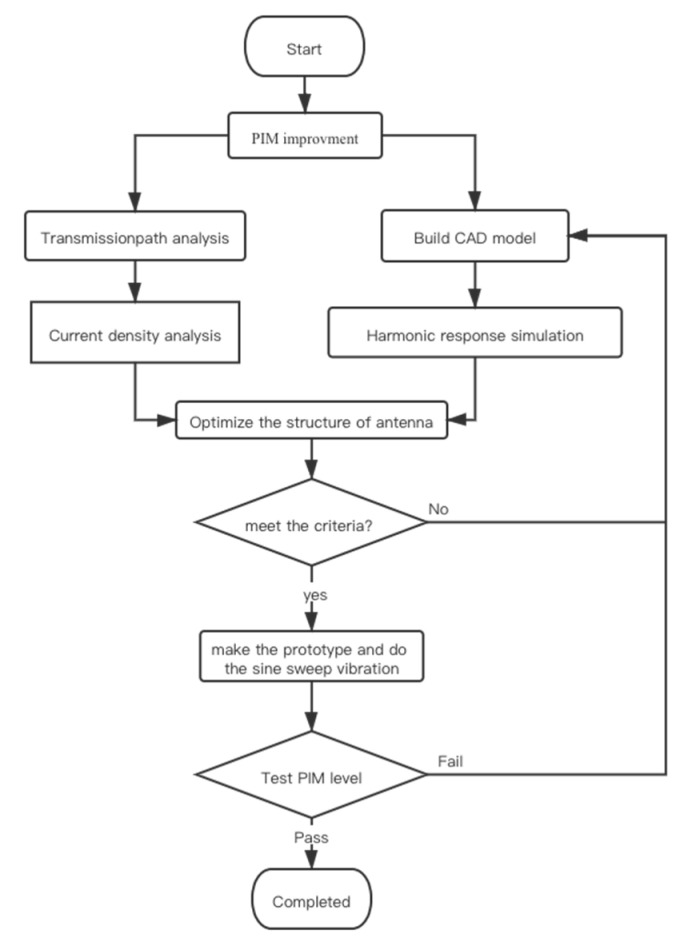
The new workflow of PIM improvement.

**Figure 7 sensors-22-00294-f007:**
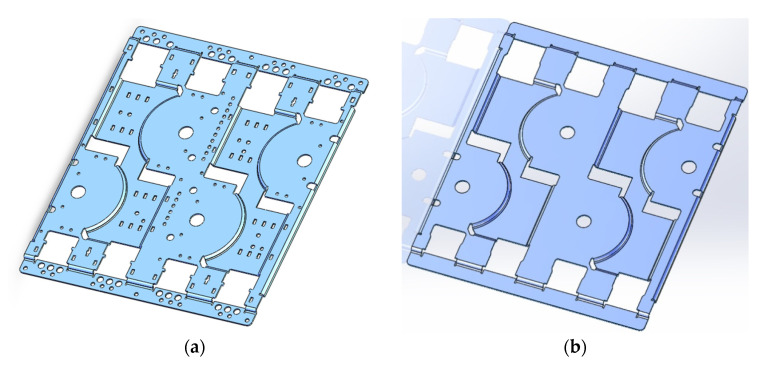
An example of simplifying the simulation model (**a**) the original model of a phase shifter plate (**b**) the simplified model of the phase shifter plate.

**Figure 8 sensors-22-00294-f008:**
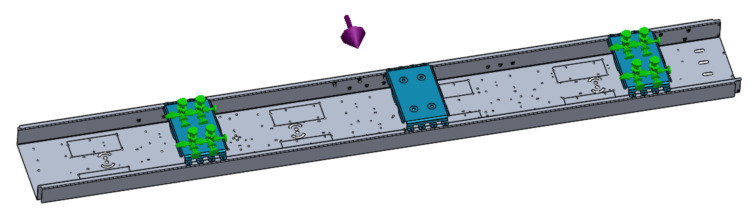
Boundary condition of simulation.

**Figure 9 sensors-22-00294-f009:**
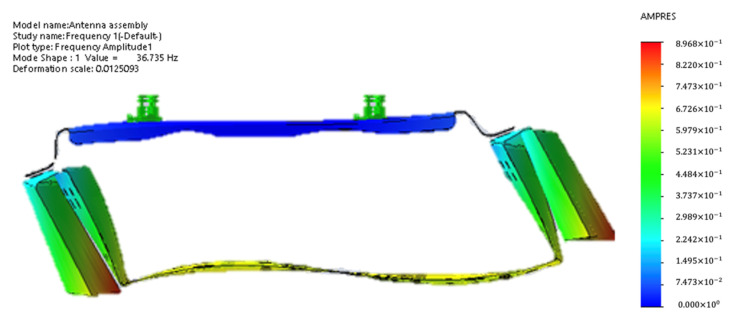
The deformation at the first-order natural frequency.

**Figure 10 sensors-22-00294-f010:**
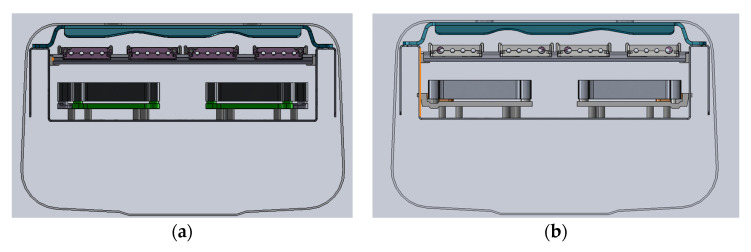
A new design to enhance the reflector: (**a**) the section view of the original design; (**b**) the section view of the improved design.

**Figure 11 sensors-22-00294-f011:**
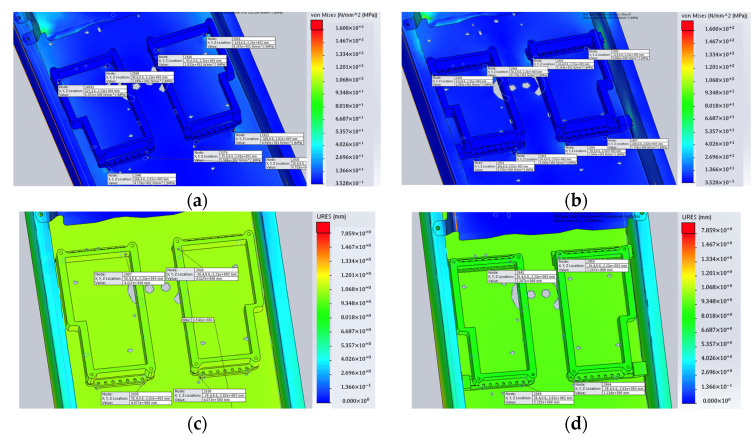
Results at crucial location between original design and improved design: (**a**) stress of original reflector at crucial location; (**b**) stress of improved reflector at crucial location; (**c**) deformation of original reflector at crucial location; (**d**) deformation of improved reflector at crucial location.

**Figure 12 sensors-22-00294-f012:**
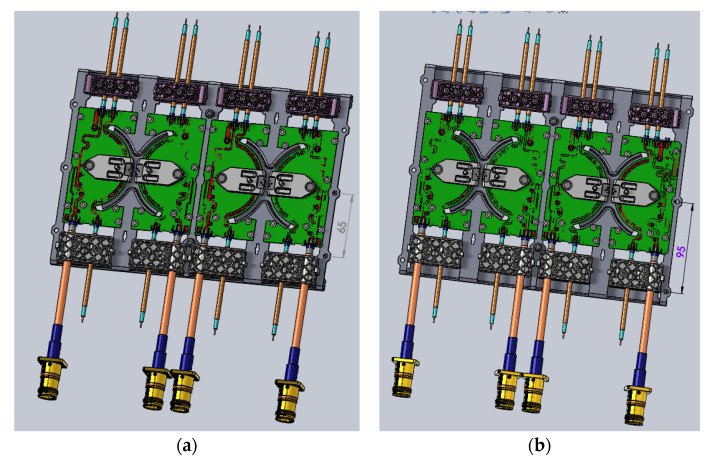
Two designs of phase shifter plate: (**a**) the original design; (**b**) the improved design.

**Figure 13 sensors-22-00294-f013:**
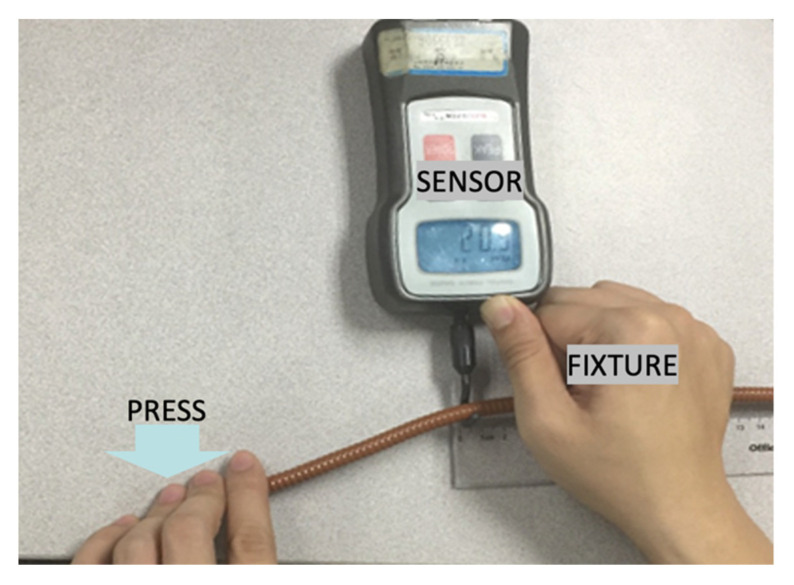
The deformation at the first-order natural frequency.

**Figure 14 sensors-22-00294-f014:**
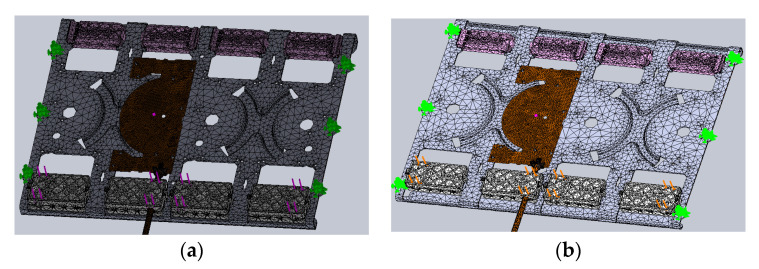
The boundary condition of the original design and the improved design: (**a**) the original design; (**b**) the improved design.

**Figure 15 sensors-22-00294-f015:**
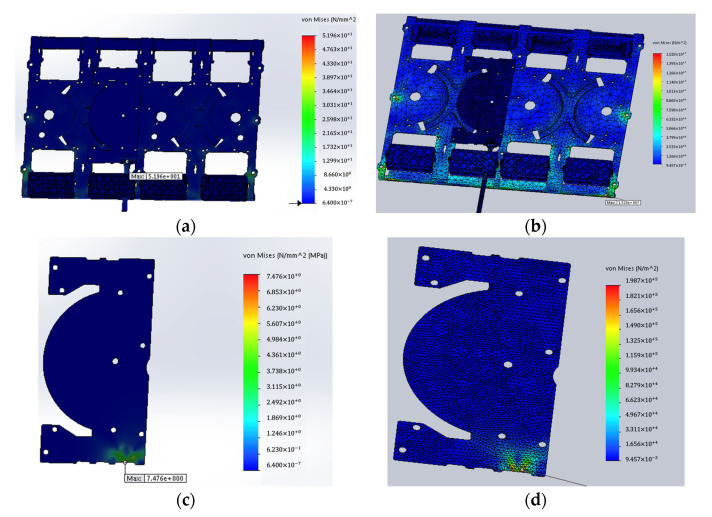
Simulation result of the original design and the improved design: (**a**) stress on the original design; (**b**) stress on the improved design; (**c**) stress on the PCB of the original design; (**d**) stress on the PCB of the improved design.

**Figure 16 sensors-22-00294-f016:**
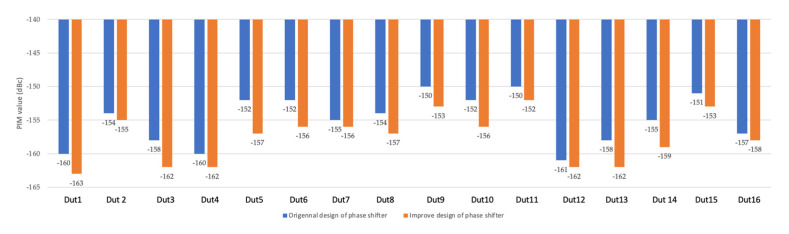
PIM test result comparison between original design of phase shifter and improved design of phase shifter after bending the input cables.

**Figure 17 sensors-22-00294-f017:**
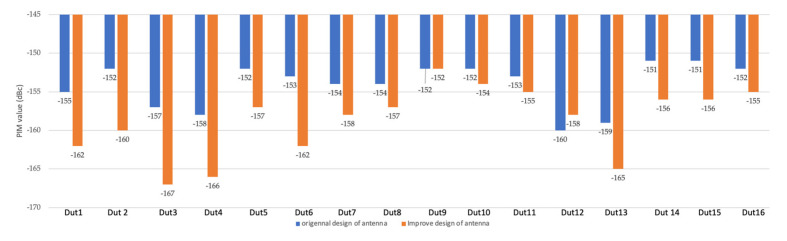
PIM test result comparison between original design of antenna and improved design antenna.

**Table 1 sensors-22-00294-t001:** Comparison result list between original design and improved design.

Item	Original Design	Improved Design	Reduction
Max stress of whole model	176.8 MPa	152.4 MPa	14%
Max stress of reflector	176.8 MPa	152.4 MPa	14%
Max deformation of reflector	5.29 mm	4.26 mm	19%

**Table 2 sensors-22-00294-t002:** Comparison results at crucial location between original design and improved design.

Item	Original Design	Improved Design	Reduction
Max stress of reflector at crucial location	82 MPa	53 MPa	35%
Max deformation of reflector at crucial location	4.12 mm	3.27 mm	21%

**Table 3 sensors-22-00294-t003:** Setting of the mechanical performance of materials.

Description of Part	Material	Young’s Modulus (GPa)	Yield Stress (MPa)	Density (kg/m^3^)
Phase shifter	PCB	20	NA	1910
Cable clamp	POM + 20%Fiber	4	150	1400
Cable connector & Inner conductor & Outer conductor	Copper	110	275.7	7400
Soldering	Tin (Sn3.5 Ag0.5Cu)	54	38	7384
Phase shifter plate	Al 5052-H32	70	195	2680

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
