# Peer review of "A Mechanical Modelling and Simulation Method for Resolving PIM Problems in Antennas"

_sensors, 2021, doi:10.3390/s22010294_

Round 1

Reviewer 1 Report

The paper proposes a new method to improve the stability of passive intermodulation (PIM) in the design phase. The method traces the possible cause of PIM and, by applying a finite element simulation of the structural design of the device under test, it optimizes its design.

What does θ = 2pi in line 153 mean?

Could the authors give a short comment on why the assumed γ3=3?

Is eq. (14) correct? I think it should be −[M]f2 + [K] = 0. Could the authors give a short proof (maybe in an appendix)?

What does exactly vertical axis in figs 14 and 15 represent? As they appear, it seems that PIM is higher in the improved design (orange bar)

Could the authors give some info regarding the finite-element simulation (e.g. equation/unknown, mesh size, running time etc)?

The paper needs moderate editing regarding the use of English.

My reviewing decision is “Reconsider after major revision”, however this mainly refers to checking the correctness of eq. (14) (due to its importance for the simulation to follows) and not to the paper as a whole.  

Reviewer 2 Report

The traditional method is to find the PIM source through measurement and then make improvements. In this paper, mechanical simulation is used to guide the design and improve the stability of the product. It is a more scientific method, and it is also meaningful for engineering. Here are some suggestions.

  1. A flowchart of the proposed method similar to Figure 1 can be provided for a clearer comparison.
  2. As far as I know, PIM is an important issue in 4G LTE base station antennas. So, are there the same PIM problems for 5G base stations, and are the solutions consistent? It can be briefly described in the introduction.
  3. Is there any data or experience that can prove that the contact at the holes in Figure 5(a) is not the main sources of PIM?
  4. Figure 7 is not very clear, there should be marks for each structure, and there should be scale labels corresponding to the colors. Figure 9 is the same.
  5. The improvement method shown in Figure 8 is not clear. The improvement should be marked, and the benefits of the improvement should be explained according to the mechanical principle.
  6. I don't quite understand the meaning of lines 244~246. Please confirm whether the statement is clear
  7. Figures 14 and 15 are not standard, the meaning of the abscissa and the ordinate should be marked and the unit should be displayed.

Round 2

Reviewer 1 Report

I am satisfied with the author's response to most of my comments.

However, regarding “point 5”, in my comment, I had meant info regarding the finite-element simulation made in this paper (e.g. equation/unknown, mesh size, running time etc) not general info on the finite-element method. Could the authors provide some relevant info?

I have also a comment on the authors’ response regarding “point 3”: I see that the authors use an equation of the form  {x}={φ}i sin(fit). However the equation should be {x}={φ}i sin(ωit) in which case the matrix equation should be –ω2[Μ] + [Κ] = 0. (I must admit that the use of f instead of ω had evaded my attention in my initial review).

I gather figs 16 and 17 (formerly figs 14 and 15) contain both, the old and the new version of the graphs.   

Author Response

Point 1: However, regarding “point 5”, in my comment, I had meant info regarding the finite-element simulation made in this paper (e.g. equation/unknown, mesh size, running time etc) not general info on the finite-element method. Could the authors provide some relevant info?

Response 1:  I add the mesh size setting principle in line 273,274, as well as running time in line 287,288.

Point 2: I have also a comment on the authors’ response regarding “point 3”: I see that the authors use an equation of the form {x}={φ}i sin(fit). However the equation should be {x}={φ}i sin(ωit) in which case the matrix equation should be –ω2[Μ] + [Κ] = 0. (I must admit that the use of f instead of ωhad evaded my attention in my initial review).

Response 2: I revised the equation 14 as your suggestion, replacing the f with ω.

Point 3: I gather figs 16 and 17 (formerly figs 14 and 15) contain both, the old and the new version of the graphs.   

Response 3: I revised the typo for the ordinals of the figure in line 454 and line 456.
